# Spontaneous, Artificial, and Genome Editing-Mediated Mutations in *Prunus*

**DOI:** 10.3390/ijms232113273

**Published:** 2022-10-31

**Authors:** Angel S. Prudencio, Sama Rahimi Devin, Sayyed Mohammad Ehsan Mahdavi, Pedro J. Martínez-García, Juan A. Salazar, Pedro Martínez-Gómez

**Affiliations:** 1Department of Plant Breeding, Centro de Edafología y Biología Apliacada del Segura-Consejo Superior de Investigaciones Científicas (CEBAS-CSIC), 30100 Espinardo, Spain; 2Department of Horticultural Science, College of Agriculture, Shiraz University, Shiraz 7144165186, Iran

**Keywords:** *Prunus*, genetic diversity, mutation, transposable elements, Ethyl Methane Sulfonate, gamma rays, transgenic plants, in vitro culture, gene editing, CRISPR, TALEN

## Abstract

Mutation is a source of genetic diversity widely used in breeding programs for the acquisition of agronomically interesting characters in commercial varieties of the *Prunus* species, as well as in the rest of crop species. Mutation can occur in nature at a very low frequency or can be induced artificially. Spontaneous or bud sport mutations in somatic cells can be vegetatively propagated to get an individual with the mutant phenotype. Unlike animals, plants have unlimited growth and totipotent cells that let somatic mutations to be transmitted to the progeny. On the other hand, in vitro tissue culture makes it possible to induce mutation in plant material and perform large screenings for mutant’s selection and cleaning of chimeras. Finally, targeted mutagenesis has been boosted by the application of CRISPR (Clustered Regularly Interspaced Short Palindromic Repeats)/Cas9 and Transcription activator-like effector nuclease (TALEN) editing technologies. Over the last few decades, environmental stressors such as global warming have been threatening the supply of global demand for food based on population growth in the near future. For this purpose, the release of new varieties adapted to such changes is a requisite, and selected or generated *Prunus* mutants by properly regulated mechanisms could be helpful to this task. In this work, we reviewed the most relevant mutations for breeding traits in *Prunus* species such as flowering time, self-compatibility, fruit quality, and disease tolerance, including new molecular perspectives in the present postgenomic era including CRISPR/Cas9 and TALEN editing technologies.

## 1. Introduction

Darwin proposed that genetic variation is vital for evolutionary adaptation. Not only is diversity important in ensuring the continuity of plant species, but also in improving product quality of a myriad of crops [1]. Additionally, genetic diversity is foremost for breeding programs, as it allows for crops to successfully adapt to environmental changes [2]. As a result, over the past few decades, a bunch of strategies have been proposed to increase genetic diversity in fruit trees [3]. One of the genetic resources that generates diversity in orchards is mutation. A mutation is defined as a sudden change in a DNA sequence not derived from segregation or recombination. It is not limited to nucleotide substitution, but to several types of changes such as deletion/insertion, inversion, and translocation [4].

The term “mutation” was coined in 1903 by Hugo de Vries in his book entitled *Die Mutations Theorie* (*Theory of mutation*) to describe the variations of phenotypes that were observed in the plant species *Oenothera lamarckiana* [5]. Mutations drive the drift of evolution and plant breeding [6]. Moreover, it is said that mutation is at the bottom of cutting-edge plant breeding techniques [3]. This phenomenon occurs in living organisms naturally through repairing genetic materials, i.e., DNA & RNA damaged by physical, chemical, and biological agents [2,7]. This term was contrived by Hugo De Vries as the abrupt substitutions in either DNA or RNA of a plant organ with no recombination or separation [5]. The derived mutation profile is called a “mutant” [8]. Mutations are the leading method in evolution [9]. They lead the modified DNA & RNA inherited by the organisms into phenotypic transformations [8]. In addition, induced genetic variability by mutation using irradiation and treatment with chemical mutagens does not imply any gene insertion, unlike genetically modified organisms (GMO) [10]. Mutation breeding has been used by plant breeders since the discovery in 1925 that heritable mutations could be induced in plants by means of irradiation or chemical treatments, and is considered as a non-transgenic crop in countries where transgenic plants are denied [11] (Figure 1).

On the other hand, conventional plant breeding techniques have developed a multitude of fruit trees for their yield and quality indicators [12]. Nonetheless, trees have a long reproductive cycle and a high degree of heterozygosity [13]. Thus, for meeting the needs of gardeners such as ameliorating the adverse impacts of abiotic stress, climate change, and global warming, or having a befitting form and phenology, e.g., dwarf, it is consequential to have a great breakthrough in the world of fruit tree breeding [12]. In terms of purpose, breeders have taken advantage of naturally occurring mutation, which has been widely used to spread an agronomical character through new varieties (e.g., late flowering or self-compatibility), and, on the other hand, developed genetically engineered material to improve fruit quality (e.g., shelf-life, color, flavor, texture, nutrient content) [12]. As an illustration, limitations such as challenges in economic, regulatory, and intellectual property hindering the progress of transgenic fruit trees are developed through mutation [14,15].

Mutations distort the double-stranded helix of DNA leading to chromosomal breaks [16]. It is reported that although some of the mutations are corrected through DNA proofreading and repair processes, some of them are transmitted into the next generation [16]. In addition, mutagenesis is a process in breeding by means of mutagens which can be inherited [3]. Therefore, it is predicted that the mutagenesis methods, i.e., insertional and site-directed mutagenesis, will be used extensively in mutation classical breeding studies owing to their precise results [2]. They can narrow juvenility in fruit crops, which is a great hindrance for breeders [5]. Moreover, the aim of shortening the stems on internodes, enhancing fruit color, developing self-fertility, and obtaining seedlessness are considered by researchers through mutation [17].

Regarding the given information about mutation, genome editing is propounded as a novel technology so as to revolutionize and expedite the genetic improvement of tree crops inside the denominated new plant breeding techniques (NPBT) [12,18]. In such a case, CRISPR (Clustered Regularly Interspaced Short Palindromic Repeats)/Cas9 and TALEN (Transcription activator-like effector nuclease) mediated gene-editing developed in bacteria are the latest technologies and accurate tools for editing genes and inducing mutations [19,20,21] (Figure 1). Nevertheless, hybrid breeding has not been expanded in the market yet, and most of the exciting fruit varieties are produced by a traditional method, i.e., clonal propagation [13].

This review will touch on spontaneous and induced mutation in *Prunus* fruit trees, including the most relevant mutations for breeding traits such as flowering time, self-compatibility, fruit quality, and disease tolerance, and the use of CRISPR/Cas9 and TALEN systems for mediated mutagenesis.

## 2. Characterization of Plant Mutations

### 2.1. Spontaneous Mutation

Spontaneous mutations, often found in somatic cells of plant bud sports, occur naturally at a very low rate (10-6%). Bud sport mutations at plastid, genic, chromosomal or genomic level are generated in a cell within a shoot apical meristem and is propagated by mitosis to the whole bud and then the other buds. This results in unstable genotypes restrained to a portion of the tree or some branches that can be propagated by clonal techniques, although the molecular base of the mutation remains unknown [4]. Such mutations can be detected and affect the structure of natural populations [22] and be useful for breeding [23]. It seems that spontaneous mutations occur more frequently in some regions of the genome, e.g., the mutation rate in GC-rich genes is more than AT-rich genes [24]. The most likely sources of spontaneous mutations are errors during DNA replication and damage from reactive oxygen species (ROS) [25]. Apart from deleterious or neutral mutations in the natural process, it is desirable to experience the mutants in the evolutionary process, thereby playing an integral role in the changing of genetic diversity and environmental conditions [26].

Natural insertions or deletions of nucleotide sequences can be due to the movement of transposable elements (TE). TE are mobile genetic factors that can account for over 50% of a plant’s nuclear genome [27]. They can be classified into two broad classes depending on their method of proliferation [28,29]. Class 1 elements are transposed by reverse transcription of RNA intermediates and cannot be removed once inserted into a new location in the genome [29]. This class contains potential retroviruses with retrotransposons and long terminal repeats (LTRs), as well as non-LTR elements [29]. LTR retrotransposons (LRNs) are the most common kind of TE [30]. Class 2 elements (also called DNA transposons) can be cut and inserted and therefore moved from one nuclear position to another [29]. Depending on the activity of the transposase encoded at another site, some coding factors mediate transposition themselves, while others are non-autonomous. TE is usually methylated via small RNA, which can extend to surrounding genes and inactivate them [31]. Insertion of TE (usually LRNs) can cause structural variation or activation of the gene, with phenotypic consequences [32]. LTR retrotransposons, some of which are of high agricultural value, are the origin of somatic mutation in plant species [33].

### 2.2. Induced Mutation

Mutations can be induced artificially in plants by agents that are classified into physical (e.g., gamma rays, X-rays, ultraviolet rays, beta particles, alpha particles, protons, fast neutrons, and ion beams), chemical (e.g., hydroxylamine, alkylating antineoplastic agents, azide, nitroso compounds, antibiotics, acridines, and base analogs), or biological (e.g., viruses and bacteria) [6,34]. Chemical and physical mutations are claimed to be more useful in plant breeding programs than other mutations, especially spontaneous ones, which are rare events, notwithstanding that a great deal of traditionally bred traits are due to natural mutations [2,6].

Mutagenesis can be performed in different plant material apart from seeds or seedlings, taking advantage of totipotency to regenerate a whole plant from an in vitro culture tissue or plant cell [4]. Frequency and types of mutations are direct results of the dosage and rate of exposure to the mutagen. Chemical mutagens are preferably used to induce single mutations, while physical mutagens induce large lesions, such as chromosomal rearrangements [35]. For instance, chemicals such as Ethyl Methane Sulfonate (EMS) basically produce single nucleotide polymorphisms (SNPs) [36]. This may change the encoded protein by premature termination and codon alterations [36]. However, polyploidy may also develop in fruit crops by chemical treatment like colchicine, which blocks mitosis without preventing DNA replication and doubles the number of chromosomes [37]. Chemical mutagens must penetrate the meristem area of the scion, and excess chemicals must be removed after the treatment [38]. Chemical mutagens are highly effective in producing optimal mutagenesis for whole plants or seeds, but they are highly toxic and are not recommended for tissue culture plants [4]. Due to this problem, mutagenesis with chemical mutagens is lower than mutagenesis with physical mutagens [39].

Physical mutagens have widely been used to induce hereditary abnormalities, and more than 70% of mutant varieties were developed using physical mutagenesis [35]. Radiation is the energy that travels in the form of particles or waves [4]. These are electromagnetic (EM) spectra with relatively high energy levels that can displace electrons from the nuclear orbitals of the atoms [4]. The term ionizing radiation was coined because the affected atoms become ions [4]. X-rays were first applied to induce mutations, and gamma rays from radioactive cobalt (^60^Co) are now widely used [4]. Although physical mutagens produce more stable genetic mutations and are less cost-effective, multisite mutations of different sizes are generated, thus affecting non-target genes [3]. For instance, using fast neutron mutagenesis results in large omission in the genome, in addition to chromosomal loss and translocations [40]. Sharafi irradiated whole almond trees using gamma rays. Late blooming, dwarf growing habit, pests, and disease tolerance were found in the 10-year study [41]. Other studies assayed irradiation of sweet cherry buds (dormant scions) to optimize dosage for recovery of useful mutations [42,43]. Finally, Yang and Schmidt cloned a mutant form X ray irradiated shoots of cherry leaves [44].

Ahloowalia [45] proposed the use of in vitro techniques for mutagenesis. Under culture conditions, a source of genetic diversity is somaclonal variation [46]. Somaclonal variation includes DNA-related epigenetic or genetic diversity that causes phenotypic changes recognizable from the parent [3]. Key causes include but are not confined to long-term in vitro culture, the presence of phytohormones, tissue culture medium composition, and other specific mechanical factors during culture [47,48]. These techniques allow a large number of proliferations for mutation induction in a small space [45]. Then, several subculture cycles can be performed in a short time to separate the mutated sectors from the non-mutated sectors [45]. The low germination rate of somatic embryos is a key obstacle to the commercialization of somatic embryogenesis [49]. Among the chemical agents, EMS is reported as a highly efficient and effective mutagen for making somaclonal variation in crops such as grapes and bananas [50]. In vitro culture can be treated with mutagens to develop established varieties [51]. Selection of culture medium with high cytokinin can cause compact mutants [51]. Additionally, somatic embryogenic culture irradiation is useful in preventing the chimeric phenomenon [52]. Otherwise, three propagation subculture cycles after irradiation are required to overcome the problem of the chimeric phenomenon [49]. Somatic embryogenesis, however, can cause rejuvenation and retard the flowering of some fruit trees [49]. Consequently, seed or budwood may be irradiated [49]. Grafting the newly developed shoots (three or four times) from the irradiated bud woods on the rootstock will help overcome the chimera problem [49].

### 2.3. Genetic Engineering Mutations

New breeding techniques such as genetic engineering integrated with mutational breeding to produce genetic diversity, can contribute to improve various characteristics in crops [53]. Genetic modification of fruit trees is not only interesting for obtaining fruit quality, stress tolerance, or for functional validation of candidate genes, but also for studying the development of non-GM scions grafted onto GM rootstocks [54,55]. The prevalent method for gene transfer is mediated by *Agrobacterium tumefaciens* infection, as this soil pathogen has the ability of deliver DNA in host plant cells. The transferred DNA (T-DNA) carries the exogenous sequences that will be integrated in the host genome. The T-DNA can contain a gene of interest to be expressed or an RNA hairpin to induce gene silencing [56].

Efficient transformation and regeneration protocols are needed for successful plant genome editing research. A low plant regeneration rate is the first bottleneck for plant genome editing [54]. Protocols for plum and apricot transformation are available [57,58], as well as for *Prunus persica* × *Prunus amygdalus* rootstocks [55,59]. Transformation has been applied for functional validation of genes such as DRO1 in plums, which is involved in shooting and rooting patterns [60]. Zong et al. [61] generated a DRO1 cherry (*Prunus cerasus* × *Prunus canescens*) overexpression mutant rootstock by Agrobacterium transformation that is able to develop adventitious root. Agrobacterium transformation has also been applied to the obtention of virus resistance, as in the case of plum resistance to *Plum Pox Virus* (PPV), and cherry rootstock resistance to *Prunus Necrotic Ring Spot Virus* (PNRSV), through gene silencing [62,63,64].

On the other hand, CRISPR/Cas9 and TALEN are currently the two systems of choice for genome editing [18,19,20,21]. CRISPR/Cas9 is a specific genome editing technique that allows the manipulation of genes accurately by deleting, adding, or modifying parts of the DNA sequence [53]. This method of inducing mutations requires two important molecules; that is, the enzyme Cas9 and a part of RNA named guide RNA (gRNA) [53]. Recently, new transformation protocols that combine Agrobacterium and CRISPR/Cas9 technologies have been released to generate transgene-free fruit apple trees and grapevine disease resistance by knocking-down target genes [56]. TALEN produced a single prevalent mutation accompanied by a variety of low-frequency mutations. The prevalent mutation was present in most tissues within a single tiller as well as in all tillers examined, suggesting that TALEN-induced mutations occurred very early in the development of plants [21].

## 3. Identification of Mutations in *Prunus* Species

In *Prunus* species, key mutations have been described as being related to most important agronomic traits including to winter dormancy and flowering time, flower self-compatibility, and fruit quality (Table 1).

### 3.1. Mutations Related to Winter Dormancy and Flowering Time

Flowering time and chilling requirement for overcoming dormancy have been shown to be polygenic traits with a high heritability in the *Prunus* genus. Different genomic regions involved in the control of chilling requirement have been identified in almond and other *Prunus* species by Quantitative Trait Loci (QTLs) [99,100,101,102]. In the almond case, a dominant gene, *Lb*, from a spontaneous late flowering bud sport of the almond variety ‘Nonpareil’ was described in Linkage Group (LG) 4, that has been used as a genetic pool for the generation of new late flowering varieties, as ‘R1000′ and ‘Tardona’ [65,66]. Another QTL region was found in LG1 in peach, in which six *DORMANCY-ASSOCIATED MADS-BOX* (*DAM*) genes tandemly arrayed are located. In this region of the *evergrowing* (*EVG*) peach mutant, which does not enter dormancy [103], four of these DAM genes are deleted [72]. In sweet cherry, mutations in *DAM* family genes were identified in an early-blooming cultivar: a large deletion (694 bp) upstream of *PavDAM1*, and various INDELs and SNPs in *PavDAM4/5* UnTranslated Regions (UTRs) [92]. Additionally, the transformation of *Arabidopsis thaliana* with peach and *P. mume MADS-box* genes produced early-flowering [73,74].

On the other hand, scanning of mutations by High-Resolution-Melting (HRM) analysis in putative floral regulators *PpAGAMOUS* and *PpTERMINAL FLOWER I* have been performed in 36 varieties of peach, detecting one polymorphic site in each gene but no phenotype association is given [75].

In apricot, the phenotype ‘Rojo Pasión Precoz’, which is an early-flowering variant is the mutation of ‘Rojo Pasión’ as the original phenotype [84]. Two early-flowering natural mutants in apricot and one in plum have been identified and may derive from their wild-relatives, as suggested by SSR and SNP fingerprinting [84]. However, the molecular basis of these mutations is unclear.

### 3.2. Mutations Related to Flower Self-Compatibility

Flower fertilization and subsequent fruit development depend on compatible pistil–pollen interactions. The compatibility system in *Prunus* species is controlled by a locus containing at least two linked genes: one expressed in the pistil that codes for an *S*-RNAse enzyme that degrades the pollen RNA in incompatible reactions; and the other in the pollen, considered to be an *S*-*haplotype-specific F-box* (*SFB*) gene [67,104].

Mutations associated to the *S*-RNase coding sequence result in low or absent RNase activity [75]. In peach, the presence of a cysteine residue instead of tyrosine as well as the cysteine residue in certain almond genotypes could be related to *S*-RNase stability [68,69]. In sour cherry, an insertion upstream of the *S*-RNase in the mutant genotype alternates a *HindIII* cut site and possibly inhibits the transcription of *S*-RNase gene [96]. Moreover, 1/23 bp deletions also conduct until disruption of the self-incompatibility system by generation of premature stop codons that are translated into truncated proteins [97]. In addition, mutations associated to the *SFB* gene expressed in pollen include partial or complete deletion of the gene or frame-shift mutations leading to premature termination of the protein in sweet cherry [93,94]. A 1 bp substitution or 100 bp deletion in *SFB* of sour cherry and a 6.8 kb insertion in the middle of *SFB* of Japanese apricot are also associated with the breakdown of self-compatibility [95,97,98]. Peach self-compatible *SFB1* and *SFB2* are mutant versions of self-incompatible alleles found in almond (by a 155 bp deletion) and Japanese plum (by a 5 bp insertion) [67]. Finally, in apricot, a 358-bp insertion is found in the *SFB* gene, resulting in the expression of a truncated protein, and thus overcoming the self-incompatibility barrier [85]. The inverse effect of the mutation of a self-compatible apricot variety into a self-incompatible variety was reported with ‘Blenheim’ apricot [86]. Self-incompatible apricot mutants (early ‘Brenham’) had larger fruits, needed less thinning, and ripened slightly earlier [17].

Interestingly, an inverted-repeat transposable element inserted into the *SFB* gene in apricot has been shown to result in self-compatibility [87]. Similarly, the presence of a gene inserted in the intergenic region between the *S*-RNase gene and *SFB* has been reported in some almond self-compatible varieties, whose expression is activated after an incompatible pollination [105].

Mutations outside of the *S* locus have also been reported to breakdown self-incompatibility in different species of *Prunus*, reviewed by Company et al. [75]. More recent examples of self-compatibility are the apricot *ParM7* gene, preferentially expressed in anthers and has a 358 bp miniature inverted-repeat transposable element (MITE) insertion [106], and the *MGST* gene in sweet cherry variety ‘Cristobalina’, with a transposon inserted in its promoter region that is associated with down-regulation of *MGST* [95].

### 3.3. Mutations Related to Fruit Quality

Generally, seedlessness induction is gained by treating apomictic seeds and scion with gamma rays [107,108], and treating seeds with thermal neutrons [76].

Flower and fruit color depend on anthocyanin and carotenoid accumulation, which can be reduced due to loss of function mutations (a 2-bp insertion or a 5-bp deletion in the third exon) of a gene coding for *GLUTATION-S-TRANSFERASE*, identified in peach [88]. The enzyme UFGT catalyzes the 3-O-glucosylation of anthocyanidins in the biosynthesis process. In Japanese apricot, up to seven nonsense mutations in the coding region of *PmUFGT3* led to the production of an inactive UFGT enzyme and thus, to the green-skinned phenotype of fruits [89]. For carotenoid accumulation, two spontaneous mutations, by retrotransposon insertion or by a TC insertion that causes a frame shift in the sequence of *CAROTENOID CLEAVAGE DIOXYGENASE-4* (*CCD4*) gene have been associated to the yellow flesh color of the fruits [90]. In this line, the knockdown of *CCD4* by Virus-Induced Gene Silencing (VIGS) renders yellow flesh phenotype as well [77]. On the other hand, Hussain et al. [78] observed loss of stiffness and increased anthocyanin accumulation of ‘Elberta’ peaches irradiated at doses between 1 and 2 kGy.

Fruit shape is affected by different mutations. It is evidenced that a long loss of heterozygosity (LOH) event may be responsible for this alteration in fruit shape [79]. Peach flat shape, which has become popular in the market worldwide, is observed when a 1.7 Mb inversion downstream of *PpOFP1* is produced. On the other hand, a ~10 Kb deletion affecting the gene *PRUPE.6G281100* co-segregates with the trait [80].

Fruit softening is affected by ethylene and indole-3-acetic acid (IAA) levels during the late-ripening stage. The stony hard phenotype is correlated with an insertion of a transposon into the 5’-flanking region of the peach *PpYUC11* gene, and the expression of this gene is low. Consistently, IAA levels were low in stony hard peaches [81]. Additionally, in peach, hairy versus glabrous fruit [82] is caused by the action of the movement of the transposon. In the Japanese plum, the variety ‘Santa Rosa’ is climacteric, and its bud sport mutant ‘Sweet Miriam’ is non-climacteric [91]. The mutant has a longer ripening period in comparison to the original cultivar; however, at the full-ripe stage, the fruits are firmer apart from the size and weight, which are the same [91]. In addition, the accumulation of sorbitol is reported to increase at the ultimate stage compared to the climacteric fruits [91]. Bud sports series of ‘Santa Rosa’ Japanese plum showed varying copy number of candidate genes related to ethylene signal transduction, thus resulting in climacteric or non-climacteric fruits [109].

Insertion of a transposon into the 50-flanking region of the peach *YUCCA* gene caused a stony-hard phenotype [81]. In addition, only in the case of peach, white versus yellow color of fruit [83], stony hard versus melting texture of the flesh [81], and hairy versus glabrous fruit [81] are caused by the action of movement of the transposon.

Regarding fruit aroma, which is partially responsible for taste and fruit quality, current studies in *Prunus persica*. Thus, the loss of function of certain *PpAAT1* mutations observed through CRISPR/Cas9 causes lower gamma-decalactone contents, which is an aromatic compound deeply linked to smell fruit [70]. Therefore, CRISPR/Cas9 is able to edit the catalyzing factor PpAAT1, an alcohol acyltransferase of aroma synthesis [53], which presents a biochemical basis for the mechanical biosynthesis of γ-decalactone and ester compounds catalyzed by PpAAT1 in peach fruits with diverse consumer preference aromas [71].

In almonds, specific *P450* genes are responsible for the bitter taste of kernels. A substitution leading to a Leu to Phe change in a *BHLH* transcription factor prevents the transcription of two *P450* genes, hence switching the taste of kernels to sweet [110]. In addition, the sweet kernel, which is necessary for the domestication of the almond phenotype, is associated with the presence of TE inserts around the *CYP71AN24 gene* [111]. This gene is involved in the synthesis of one of the major enzymes of the amygdalin pathway (cytochrome P450), and it has been proposed that the lack of expression of *CYP79D16* along with its reduced expression results in sweet kernel properties [112].

## 4. New Molecular Perspectives in the Postgenomic Era

At this moment, in the present postgenomic era, different *Prunus* species clustered in five fruit groups including almond have been sequenced and their reference genomes are available in the Genome Database for Rosaceae (Table 2) including peach (*P. persica*) and related species (*P. kansuensis*, *P. davidiana*, *P. ferganesis*, and *P. mira*), apricot (*P. armeniaca*) and related species (*P. madshurica* and *P. sibirica*), plum (*P. salicina* P, prune (*P. domestica*), mei (*P. mume*), sweet cherry (*P. avium*), almond (*P. dulcis*), and other species such as *P. yedoensis* and *P. humilus* [113].

This whole sequenced work started in 2002 with the development of the reference genome of the peach. The development of complete genomes is making any organism accessible and amendable for many kinds of studies which will allow a precise reference of the molecular results obtained, and the development of high-throughput methods for genomic analysis involving the most abundant genetic variation (Single Nucleotide Polymorphisms, SNPs) and transcriptomic analysis at differential gene expression (DEG) level in a new postgenomic perspective. At this moment, the new Big Data Biology harboring the development of DNA (and RNA as cDNA) high-throughput sequencing technologies together with bioinformatics analysis as well as the creation of databases with billions of data has made possible to access genetics knowledge at the level of each nucleotide. This new methodological perspective where millions of sequences are available in one single experiment with the detailed information of complete genomes (defined as the DNA organized into separate chromosomes inside the nucleus of a cell) and transcriptomes (described as the complete list of all types of RNA molecules). Several authors have even characterized this data-intensive biology as a new kind of science, a science of information management, different from traditional biology. Big data science is now being introduced in the development of molecular markers to assist breeding selection methodologies (Table 2).

This new postgenomic perspective integrating available reference genomes and new sequencing and bioinformatic methodologies will allow the implementation of new Marker Assisted Selection (MAS) to accelerate the breeding process and the use of identified and induced *Prunus* mutations. High-throughput sequencing technologies resulted in a great advance in the development and application of MAS strategies. This situation does not mean that MAS will replace conventional breeding; however, other ways are a necessary complement. In this context, there has been a significant shift of conventional breeding to molecular breeding. However, in both conventional and molecular breeding, the target must be well defined in a complete integrated work plan to accelerate the incorporation of desirable traits such as self-compatibility, frost resistance, disease resistance, or fruit quality, also including the possibilities of use of identified or induced mutations.

In addition, in this postgenomic context, editing technologies such as CRISPR/Cas9 and TALEN appears as a new strategy for plant breeding, outpacing the slow traditional methods. The ease of its application for genetic manipulation in a wide range of plants, allows scientists to establish genetic diversity, improving crop production, studying nutritional benefits, as well as disease tolerance. In addition, the discovery of several Cas9 orthologs increases the versatility and efficiency of the CRISPR and TALEN editor system, which favors being able to cover different breeding programs [132]. Therefore, CRISPR/Cas9 and TALEN technologies open a new era in plant breeding by gene editing. The use of these technologies could accelerate the knowledge of the biological process involved in any trait of interest, such as fruit quality, postharvest performance, or abiotic and biotic stresses, also including the use of the identified mutations in the case of *Prunus* species. However, to date, very few contributions have been made in *Prunus* species. The first studies using CRISPR gene editing technology were oriented towards model plant species such as Arabidopsis and tobacco [133].

In the case of important diseases such as *Monilinia fructicola* fungus, in *Prunus* species some studies are being focused. To date, pathogenic and genomic studies are scarce in this disease, but recent advances have identified a *redox-related gene* (*MfOfd1*) being up-regulated after its inoculation [134]. In this work, CRISPR/Cas9 was used combined with homologous recombination in order to determine the role of MfOfd1 gene. The results showed a sporulation decrease in the knockdown transformants at least of 50% reaching at best 80% which was linked to lower fungal virulence on peach fruit compared to wild type. A similar approach was implemented for the *Lasiodiplodia theobromae* fungus, which is one of the main causal agents of peach gummosis, where the LtAP1 gene has been reported as up-regulated in peach shoots during the infection, being this gene characterized by CRISPR-Cas9 system and homologous recombination [135]. The results appointed that a LtAP1-deletion mutant was responsible for slower vegetative growth.

Another application of CRISPR/Cas9 enrichment has been through the combination with long-read sequencing technology in order to solve complex genomic regions. This has been successfully applied to the *MYB10* region on chromosome 3, which is significantly linked to fruit color variability in *Prunus* species [136]. In this work, it was possible to apply this technology to cleave *MYB10* genes of five plum varieties (*Prunus salicina* L.) proving to be a useful tool for long-read targeted sequencing when the reference genome was not available.

On the other side, in crops as important as wheat, the highly efficient gene edition mediated by TALEN has been achieved [137]. In this assay, a TALEN pair which was targeting the *uidA* gene (*E. coli*) was co-transformed into embryos from a wheat Fielder cultivar. As a result, twelve lines were obtained containing both TALEN and *uidA* copies. Finally, they sequenced PCR products obtained from three plants, identifying different deletions that destroyed the BclI site, with this enzyme being responsible for cleaving the target site.

In *Prunus*, any TALEN application has been described to date, although Ilardi and Tavazza [138] highlighted the interest of this biotechnological strategy as a tool for *Plum pox virus* resistance in these species. However, a successful application of this genome editing tool has been described in other fruit trees such as apple [139] and fig [140].

Finally, regarding regulatory approaches, applications of genome editing tools (CRISPR and TALEN) necessitate the generation of a new organism considered in general as genetically modified organism (GMO) [21,141] and so would be captured by GMO legislation and liable for risk assessment, authorization, and in some countries mainly in Europe, product labelling. In Europe, crops designated as GMOs are much more strictly regulated than in the Americas and Asia. These GMOs are covered by Regulation (EC) no. 1829/2003 on genetically modified food and feed and Directive 2001/18/ EC on the release of GMOs into the environment [142]. The exact procedures vary between the competent authorities in different countries, but many are costly and have long time frames [37]. Several countries in North and South America, along with others in Asia, have, however, ruled that products of simple gene editing are not GMOs and would be regulated as any other conventionally bred variety [37]. US government administration allows one edit (be it a double-strand break or a nucleotide) to be made. Edited gene families and homeologs are regulated articles (US does not use the term GM), and not conventional. In addition, the US is unique in that it now allows cisgenes to be considered conventional, even if transferred by genome editing. An indel will be allowed as conventional as long as an allele with the exact nucleotide sequence exists in a sexually compatible relative [37].

In Europe, prior to the European Court of Justice (ECJ) ruling on mutagenesis in 2018, there was a general expectation that the EU would follow a similar path by treating gene-edited mutations in a similar manner to classical mutagenesis with both being exempted from EU GMO legislation. Technically, mutation breeding was defined in Directive EC 2001/18 as a technique of genetic modification but exempted from the regulation because it was considered, at the time, to have a history of safe use. However, the ECJ ruled that gene editing and other new forms of mutagenesis did not fit the exemption and that even simple mutations generated by gene editing must be regulated as GMOs [141].

## 5. Conclusions

Genetic diversity relies on the abundance and variation of alleles among individuals in a population. It is part of the evolutionary process that enables organisms to adapt to changing conditions. Populations with high allelic diversity easily adapt to abiotic and biotic stresses. Nevertheless, presently, the permanent application of populations with specific characteristics in plant breeding and the monotony of consumers’ demands are some of the reasons for genetic erosion. The loss of genetic variation of a species may lead to the loss of beneficial properties for humans. In the event of stressful conditions such as drought and disease, a population’s capability to survive by adapting to these new conditions depends on the presence of individuals with gene alleles that must adapt to these conditions. The long juvenile phase, perennial nature, sexual incompatibility, heterozygotes, etc., in fruit crops restricts their amelioration through traditional breeding. Mutations may be made artificially with the assistance of different physical and chemical agents called mutagens. The most common chemical and physical mutagens are EMS and gamma rays, respectively. Both spontaneous and induced mutations have the potential to boost new varieties matching the breeding of floral and fruit traits, and the resistance or tolerance of the fruit tree to biotic or abiotic stresses, as virus infection. In the present postgenomic era, different *Prunus* species have been sequenced with the availability of dozens of refence genomes to accelerate the breeding process and the use of identified and induced *Prunus* mutations. In addition, genome editing techniques, especially CRISPR/Cas9 and TALEN, promise to be more effective and accurate in gene editing if the genome sequence of the target gene is known. CRISPR/Cas9 and TALEN technologies open a new era in plant breeding by gene editing, which could accelerate the related knowledge with any biological process involved in any trait of interest, such as fruit quality, postharvest performance, or abiotic and biotic stresses, also including the use of identified mutations in the case of the *Prunus* species. Genome editing-mediated mutations conform to a relatively rapid method of generating characterized mutations in target genes, and should be advantageous from a regulatory perspective in that new varieties would not require expensive GMO authorization or labelling.

## Figures and Tables

**Figure 1 ijms-23-13273-f001:**
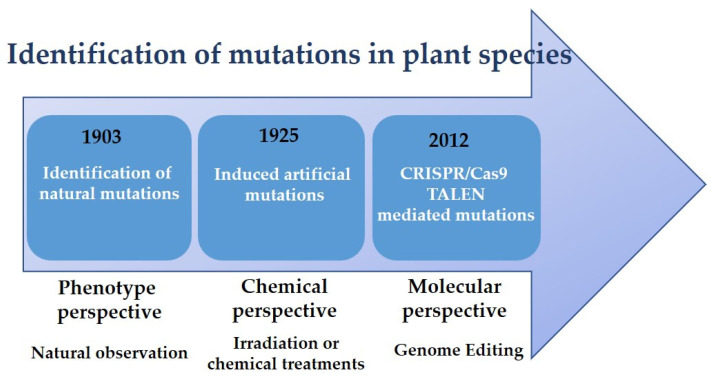
Historical perspective of the identification of mutations in plant species.

**Table 1 ijms-23-13273-t001:** Identified mutations related to most important agronomic traits in *Prunus* species.

Species	Winter Dormancy and Flowering	Flower Self-Compatibility	Fruit Quality
Subgenus *Amygdalus*			
Section Euamygdalus (almond and peach group)			
Almond (*P. dulcis)*	[65,66]	[67,68,69]	[70,71]
Peach (*P. persica*)	[72,73,74,75]		[76,77,78,79,80,81,82,83]
Subgenus *Prunus*			
Section Armeniaca (apricot group)			
Apricot (*P. armeniaca*)	[84]	[17,85,86,87]	
Mei (*P. mume*)	[73,74]		[88,89,90]
Section Prunus (plum group)			
Plum (*P. salicina*)		[68]	[82,91]
Subgenus *Cerasus*			
Section Microcerasus (cherry group)			
Sweet Cherry (*P. avium)*	[92]	[93,94,95]	[72]
Sour Cherry (*P. cerasus*)		[95,96,97,98]	

**Table 2 ijms-23-13273-t002:** Summary of the *Prunus* Genomes data available in the Genome Database for *Rosaceae* (https://www.rosaceae.org/tools/jbrowse, accessed on 28 September 2022).

Species Group	Genome	Reference
Peach	*Prunus persica* cv. Lovell. Genome v1.0	Verde et al., 2013 [114]
	*Prunus persica* cv. Lovell. Genome v2.0.a1	Verde et al., 2017 [115]
	*Prunus persica* cv. 124. Pan. Genome v1.0	Zhang et al., 2021 [116]
	*Prunus persica* cv. Zhongyoutao 14. Genome v1.0	Lian et al., 2021 [117]
	*Prunus persica* cv. Chinese Cling. Genome v1.0	Cao et al., 2021 [118]
	*Prunus kansuensis*. Genome v1.0	Submitted for publication
	*Prunus kansuensis*. Genome v2.0	Cao et al., 2021 [118]
	*Prunus davidiana*. Genomne v1.0	Submitted for publication
	*Prunus davidiana*. Genomne v2.0	Cao et al., 2022 [119]
	*Prunus ferganensis*. Genome v1.0	Submitted for publication
	*Prunus ferganensis*. Genome v2.0	Cao et al., 2022 [119]
	*Prunus mira*. Genome v1.0	Submitted for publication
	*Prunus mira*. Genome v2.0	Cao et al., 2022 [119]
Apricot	*Prunus armeniaca* cv. Chuanzhihong and Dabaixing. v1.0	Jiang et al., 2019 [120]
	*Prunus armeniaca* cv. Marouch n14. Whole Genome v1.0	Groppi et al., 2021 [121]
	*Prunus armeniaca* cv. Stella. Whole Genome v1.0	Groppi et al., 2021 [121]
	*Prunus armeniaca* cv. Sungold. Whole Genome v1.0	Submitted for publication
	*Prunus armeniaca* cv. Longwangmao. Whole Genome v1.0	Submitted for publication
	*Prunus mandshurica* cv. CH264_4. Whole Genome v1.0	Groppi et al., 2021 [121]
	*Prunus sibirica* cv. CH320_5. Whole Genome v1.0	Groppi et al., 2021 [121]
	*Prunus sibirica* cv. F106. Whole Genome v1.0	Submitted for publication
Plum	*Prunus salicina* cv. Sanyueli Genome v2.0	Liu et al., 2020 [122]
	*Prunus domestica*. Draft Genome v1.0.a1	Callahan et al., 2021 [123]
	*Prunus salicina* cv. Zhongli No. 6 Genome v1.0	Huang et al., 2021 [124]
	*Prunus mume* cv. Tortuosa. Genome v1.0	Zheng et al., 2022 [125]
	*Prunus salicina* cv. Sanyueli. Genome v1.0	Fang et al., 2022 [126]
Cherry	*Prunus avium* cv. Satonishiki. Genome v1.0.a1	Shirasawa et al., 2017 [127]
	*Prunus avium* cv. Tieton. Genome v1.0.a1	Wang et al., 2020 [128]
	*Prunus avium* cv. Tieton. Genome v2.0	Wang et al., 2020 [128]
Almond	*Prunus dulcis* cv. Lauranne. Genome v1.0	Sánchez-Pérez et al., 2019 [110]
	*Prunus dulcis* cv. Texas. Genome v2.0	Alioto et al., 2020 [111]
	*Prunus dulcis* cv. Nonpareil. Genome v2.0	D’Amico et al., 2022 [129]
Other	*Prunus yedoensis* cv. *P. pendula* f × *P. jamasakura.* Genome v1.0	Baek et al., 2018 [130]
	*Prunus humilus*. Genome v1.0.a1	Wang et al., 2022 [131]

## Data Availability

Not applicable.

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
