# Peer review of "Spontaneous, Artificial, and Genome Editing-Mediated Mutations in Prunus"

_ijms, 2022, doi:10.3390/ijms232113273_

Round 1

Reviewer 1 Report

This article reviews breeding in Prunus species and discusses the potential of new genetic breeding techniques. I think the topic is interesting, but the manuscript needs some editing. There are many sentences that are too long and need clarification. Also, there needs to be better transitions between topics and thoughts. Two topics that I think could be elaborated on are: 1) Different genetic modification techniques (TALEN was brought up towards the end of the manuscript, but this could be presented early)  (2) Potential challenges of implementing CrispR (Societal acceptance?). See my other editorial comments in the attached PDF. 

Author Response

October 27, 2022

Dear Mr. Jesse Yi, Assistant Editor of International Journal of Molecular Sciences,

please find enclosed the manuscript ijms-1973665-R1 entitled “Spontaneous, Artificial and Genome Editing-mediated Mutations in Prunus” which is the revised version of the former manuscript “Spontaneous, Artificial and CRISPR/Cas9-mediated Mutations in Prunus”, which we would like to publish in your journal.

According with the suggestions of the Reviewer 1 we have revised the manuscript incorporating the proposed revisions indicating these revisions with the control of changes of the WORD document.

We deeply appreciate the efforts of the reviewer in the improvement of the manuscript for a future publication.

Regarding reviewer's 1 comments (R1): 

R1: This article reviews breeding in Prunus species and discusses the potential of new genetic breeding techniques. I think the topic is interesting, but the manuscript needs some editing. 
Authors: We agree and thank the Reviewer 1 for their comments about the revision of this work and for considering this manuscript suitable for future publication. In addition, all the suggestions and revisions of the reviewer have been incorporated indicating these revisions with the “Track Changes” of the WORD document. 

R1: There are many sentences that are too long and need clarification. Also, there needs to be better transitions between topics and thoughts.
Authors: We agree and thank the Reviewer 1 for their comments about the edition of the manuscript. The mentioned sentences in the PDF file have been rewritten to improve the comprehension of the manuscript. English language and style were checked with the help of a English spoken college and improved. 

R1: Two topics that I think could be elaborated on are: 1) Different genetic modification techniques (TALEN was brought up towards the end of the manuscript, but this could be presented early) (2) Potential challenges of implementing CrispR (Societal acceptance?). 
Authors: We agree and thank the Reviewer 1 for their comments about the addition of some discussion about TALEN and social acceptance of editing technologies. These new comments have been included around the whole manuscript and mainly in the section 4 New molecular perspectives in the postgenomic era in pages 10 and 11. In this context, Figure 1 has been revised. In addition, 7 new references regarding these topics have been added. 

R1:. See my other editorial comments in the attached PDF.
Authors: We agree and thank the Reviewer 1 for their editorial comments. All the indicated editorial comments in the attached PDF around the whole manuscript have been incorporated in the revised version of the manuscript. 

We deeply again appreciate the efforts of the reviewer in the improvement of the manuscript for a future publication. This acknowledgement has been incorporated to the Acknowledgments section in page 12.

Yours faithfully,

Dr. Pedro Martínez-Gómez
CEBAS-CSIC, Murcia (Spain)

Reviewer 2 Report

This article comprehensively summarized the latest progress in spontaneous mutation, artificial mutation and gene editing of prunus plants, and provided a useful reference for genetic improvement of Plum plants. The article has comprehensive references and fluent language. It is recommended to receive and publish in IJMS.

Author Response

Dear Mr. Jesse Yi, Assistant Editor of International Journal of Molecular Sciences,

please find enclosed the manuscript ijms-1973665-R1 entitled “Spontaneous, Artificial and Genome Editing-mediated Mutations in Prunus” which is the revised version of the former manuscript “Spontaneous, Artificial and CRISPR/Cas9-mediated Mutations in Prunus”, which we would like to publish in your journal.

According with the suggestions of the Reviewer 2 we have revised the manuscript incorporating the proposed revisions indicating these revisions with the control of changes of the WORD document.

We deeply appreciate the efforts of the reviewer in the improvement of the manuscript for a future publication.

Regarding reviewer's 2 comments (R2): 

R2: This article comprehensively summarized the latest progress in spontaneous mutation, artificial mutation and gene editing of prunus plants, and provided a useful reference for genetic improvement of Plum plants. The article has comprehensive references and fluent language. It is recommended to receive and publish in IJMS.
Authors: We agree and thank the Reviewer 2 for their valuable comments about the interest of this work. In addition, some revisions have been performed to improve the comprehension of the manuscript attending the interest showed by Reviewer 2 and the comments of Reviewer 1. These suggestions and revisions of the reviewer have been incorporated indicating these revisions with the “Track Changes” of the WORD document. 

We deeply again appreciate the efforts of the reviewer in the improvement of the manuscript for a future publication. This acknowledgement has been incorporated to the Acknowledgments section in page 12. 

Yours faithfully,

Dr. Pedro Martínez-Gómez
CEBAS-CSIC, Murcia (Spain)